# Unsupervised Video Anomaly Detection Based on Similarity with Predefined Text Descriptions

**DOI:** 10.3390/s23146256

**Published:** 2023-07-09

**Authors:** Jaehyun Kim, Seongwook Yoon, Taehyeon Choi, Sanghoon Sull

**Affiliations:** School of Electrical Engineering, Korea University, Seoul 02841, Republic of Korea; jhkim@mpeg.korea.ac.kr (J.K.); swyoon@mpeg.korea.ac.kr (S.Y.); taehyeon@korea.ac.kr (T.C.)

**Keywords:** abnormal video, CLIP, embedding space, fine-tuning of pre-trained models, large language models, large vision and language models, similarity measure, text descriptions, unsupervised video anomaly detection

## Abstract

Research on video anomaly detection has mainly been based on video data. However, many real-world cases involve users who can conceive potential normal and abnormal situations within the anomaly detection domain. This domain knowledge can be conveniently expressed as text descriptions, such as “walking” or “people fighting”, which can be easily obtained, customized for specific applications, and applied to unseen abnormal videos not included in the training dataset. We explore the potential of using these text descriptions with unlabeled video datasets. We use large language models to obtain text descriptions and leverage them to detect abnormal frames by calculating the cosine similarity between the input frame and text descriptions using the CLIP visual language model. To enhance the performance, we refined the CLIP-derived cosine similarity using an unlabeled dataset and the proposed text-conditional similarity, which is a similarity measure between two vectors based on additional learnable parameters and a triplet loss. The proposed method has a simple training and inference process that avoids the computationally intensive analyses of optical flow or multiple frames. The experimental results demonstrate that the proposed method outperforms unsupervised methods by showing 8% and 13% better AUC scores for the ShanghaiTech and UCFcrime datasets, respectively. Although the proposed method shows −6% and −5% than weakly supervised methods for those datasets, in abnormal videos, the proposed method shows 17% and 5% better AUC scores, which means that the proposed method shows comparable results with weakly supervised methods that require resource-intensive dataset labeling. These outcomes validate the potential of using text descriptions in unsupervised video anomaly detection.

## 1. Introduction

At present, surveillance cameras are being extensively used to monitor abnormal situations to ensure the security and safety of individuals and properties. To lessen the labor of monitoring workers, there is a significant amount of research on automatic video anomaly detection algorithms. With the advent of deep learning approaches, the performance of anomaly detection methods increases by a large margin. Since deep learning approaches mainly depend on the dataset, various anomaly detection methods are also dependent on the assumption of dataset configuration, such as the existence of abnormal videos or labels for the dataset. Each configuration reflects the cost of labeling each frame and the difficulty of obtaining abnormal videos in the real world.

According to the dataset configuration, the approaches for anomaly detection in video surveillance can be categorized into one-class classification, weakly supervised, and unsupervised techniques, following the terms used in GCL [1]. The one-class classification approach uses a training dataset consisting of only normal situations [2,3,4,5,6,7,8,9,10]. In the weakly supervised approach, video-level labeled abnormal data are used for training, in addition to the normal data used in the one-class classification approach [11,12,13,14,15,16,17]. The unsupervised approach uses an unlabeled training dataset, most of which consists of normal situations [1,18,19]. Notably, both the one-class classification and unsupervised approaches assume that all or most of the training video data are normal, which limits their effectiveness in discriminating situations that are not frequently encountered in the training dataset. Although the weakly supervised approach attempts to address this problem by incorporating additional abnormal video clips for training, there remains a possibility of encountering abnormal situations that are not included in the training dataset.

These challenges can be overcome by acquiring a sufficient amount of normal and abnormal training video data that cover various situations. However, the acquisition and labeling of such video data is expected to be cost-intensive. Instead of relying solely on labeled video data, it may be preferable to use text descriptions of potential normal and abnormal situations, such as “riding a bike” and “vehicle accidents”, because the users within a specific application of anomaly detection typically have knowledge pertaining to the possible normal and abnormal situations. These text descriptions can provide insights into normal and abnormal situations that may not be present or labeled in the training video dataset.

Recently, an abnormal human action recognition method [20] that uses text descriptions has been proposed. It relieves the cost of gathering abnormal human datasets by using abnormal text descriptions. Similarly, we try to utilize text descriptions, but we apply these text descriptions in the video anomaly detection fields, which has a much broader sense of anomaly situations than abnormal human action recognition. Video anomaly detection can cover situations that do not contain human actions, such as explosions, rockslides, and the occurrence of various abnormal objects.

Thus, the objective of this study is to explore the potential of using a suitable set of texts describing normal and abnormal videos to reduce the costs of data gathering and labeling. We assume that the unlabeled video dataset is available, which mainly consists of normal situation data, reflecting the rarity of abnormal events and presence of noise in real-world scenarios. With this video dataset, we attempt to verify the feasibility of using these text descriptions in unsupervised anomaly detection. First, we obtain text descriptions for the target application of anomaly detection. Specifically, we use one of the large language models (LLMs), ChatGPT [21], to generate the text descriptions, similar to the generation method in [22]. In this manner, we obtain text descriptions that can be applied to the general domain of anomaly detection. Subsequently, we modify the descriptions by considering specific domains of anomaly detection, such as a strict pedestrian-only zone in which a running person, car, or motorbike is not allowed.

Next, we apply these predefined text descriptions to anomaly detection. We use one of the large vision and language models (VLMs) [23,24,25,26,27,28,29], i.e., the contrastive language–image pre-training (CLIP) model [23], which can effectively find similarities between images and texts. We detect abnormal situations based on similarities adequate enough to discriminate between normal and abnormal situations in surveillance videos. However, one can raise the issue that only one image is not enough to obtain enough information about the temporal dependency. For this reason, various existing methods use optical flow [6,9,30] or multiple frames [1,12] to capture the high-level motion which is real motion in the scene, such as the movement of people when fighting. We assume that one image is enough to capture this high-level information with the aid of VLMs, as shown in Figure 1. The figure shows the top five most similar images for each text description, determined based on the cosine similarity of the image and text features of CLIP. For the ShanghaiTech dataset [31], we use cropped images using an object detector [32]. The image and corresponding text feature of CLIP are remarkably similar, especially for the objects and images containing high-level motion information, such as a photo of people engaged in a fight and another capturing a road accident.

By leveraging the capabilities of CLIP [23], we can effectively detect abnormal frames within the video. To enhance the performance of anomaly detection in the target domain, we use an unlabeled video dataset to adjust the similarities between the image and text features in CLIP. Several approaches for a classification task have been developed for adapting large VLMs to the target domain with a labeled dataset [33,34,35,36,37,38]. However, we use only the unlabeled video dataset for the adaptation to ensure that the similarities align with the target domain, resulting in further enhancement in the anomaly detection performance.

We hypothesized that the adaptation to the unlabeled video dataset could be achieved by increasing or decreasing the similarity score for each text description according to the frequency of the situation in the dataset. For example, we can perform adaptation by increasing the similarity score with “car”, because it is frequently observed in the training dataset. To reflect this motivation, we introduce a similarity measure, termed *text-conditional similarity*, which has two kinds of trainable parameters, both of which can increase or decrease the similarity score with text descriptions intuitively. The first parameter gives a weight to each coordinate component when calculating the cosine similarity. The second one is a scalar value that can directly adjust the similarity score for the corresponding text description.

To train the parameters, we employ triplet loss to magnify the difference between the proposed similarity scores for normal and abnormal text descriptions. Additionally, we introduce a regularization loss that takes into account prior knowledge pertaining to the unsupervised dataset, in which most of the data correspond to normal scenarios. After training the parameters, we can easily detect abnormal situations in videos by calculating the proposed text-conditional similarity between the input image feature and text features. The training and inference processes are straightforward and the proposed methods do not necessitate the computationally expensive calculation of optical flows [9] or feature extraction from multiple frames [12], which are typically required by other anomaly detection methods.

The aforementioned work, the abnormal human action recognition method [20], also leverages CLIP [23] to utilize the text descriptions. In addition to the difference we mentioned above, this work differs from the proposed method in three aspects. The first difference pertains to the dataset scenario. The existing approach requires action class labels for pre-training the deep feature of the 2D skeleton and uses a normal-only dataset. The second difference pertains to the computational complexity for training. The existing approach does not use an image feature extractor in CLIP and instead trains a feature extractor from scratch using labeled data. Consequently, this method requires a considerable amount of time for training, unlike the proposed method. The third difference pertains to the abnormal situations and corresponding prompts. The existing approach uses only five prompts related to situations involving a fight. In contrast, the proposed approach uses a large number of prompts to address various abnormal situations.

We use two datasets, i.e., the ShanghaiTech [31] and UCFcrime [11]. The proposed method outperforms the existing video-based unsupervised methods and achieves results that are comparable to those of weakly supervised methods that typically involve the labor-intensive task of labeling normal and abnormal videos.

The contributions of our work can be summarized as follows:We propose a simple unsupervised video anomaly detection method based on the improved similarity measure, text-conditional similarity, between the input frame and pre-defined multiple text descriptions of normal and abnormal situations using CLIP.We propose a training process for the proposed similarity measure using triplet and regularization loss without labels.We leverage LLMs to easily obtain text descriptions for normal and abnormal situations.We demonstrate the feasibility of the proposed approach by comparing it with the existing methods. The proposed method outperforms the existing unsupervised methods and achieves results that are comparable to those of weakly supervised methods, which incur additional costs for data labeling.

The remaining paper is organized as follows. Section 2 reviews the related work. Section 3 describes the proposed anomaly detector using text descriptions. Section 4 presents the experimental results. Section 5 presents the concluding remarks.

## 2. Related Work

This section summarizes the existing methods for anomaly detection in surveillance video, large VLMs, and applications of large VLMs to specific domains.

### 2.1. Anomaly Detection in Surveillance Video

As discussed in the Introduction, anomaly detection in surveillance video can be categorized into three types based on the training data: one-class classification (OCC), unsupervised, and weakly supervised, following the terms used in GCL [1]. The first category assumes that only normal data are available for training. This assumption has been used from the classical approaches that do not use deep learning methods. They use conventional features, such as histogram of gradient (HoG) or histogram of optical flow (HOF) [2,3], 3DSIFT [4], etc. After the advent of deep learning methods, deep autoencoders that reconstruct the image [7] or optical flow [5,6] are widely used to capture the normal patterns of appearance and motion in the video. To reflect the temporal dependency to the autoencoder, some methods use prediction task to train the autoencoder [8]. Furthermore, some methods use pseudo-abnormal data to make up for the absence of abnormal data [7].

Recently, an OCC method that shows promising anomaly detection results has been proposed by Georgescu et al. [9]. Their method belongs to object-centric anomaly detection, which performs anomaly detection for each detected object from an object detector. When at least one detected object is determined as abnormal, they determine that abnormal situations occur in the frame. They use two kinds of reconstruction networks, one for detected objects and the other for the optical flow. They use pseudo-abnormal data from various datasets such as [39], flower images [40], and anime images, to make up for the absence of abnormal data. In the experiment, we train the model using an unlabeled dataset that contains a small portion of abnormal data. When applied to the unlabeled dataset, they show even better results than the existing SOTA unsupervised method because their reconstructor is mainly trained on normal training data. However, their method requires optical flow estimation for forward and backward optical flow in the inference process which can be a computational burden. Furthermore, their method is object-centric, so it cannot be applied to abnormal situations that a typical object detector cannot detect, such as explosion and arson in the UCFcrime dataset [11]. Another effective object-centric anomaly detection method by Reiss et al. [10] shows even better results than Georgescu et al.’s [9]. They calculate a feature from optical flow, 2D skeleton, and deep feature from CLIP [23] for detected objects in the training dataset. In the inference, they estimate density using a Gaussian mixture model (GMM) for optical flow and *k*NN with the training dataset for 2D skeleton and deep feature. However, *k*NN distances from Reiss et al. [10] cannot be used as an unsupervised approach, because an abnormal feature in training data makes *k*NN distance small, which makes the anomaly score for abnormal data become small. All of these OCC approaches require normal-only training dataset and, due to the absence of abnormal data, their performance is limited.

Approaches in the second category incorporate abnormal-labeled videos along with normal data during training, which is a weakly supervised method. The dataset is assumed to have two kinds of video, normal and abnormal. The normal videos have normal frames only. The abnormal videos have both normal and abnormal frames but are not labeled by frame level. Sultani et al. [11] first posed the dataset scenario of the weakly supervised approach by publishing the UCFcrime dataset [11]. They used a classifier that was trained to output an anomaly score that is zero for a normal and one for an abnormal frame. They trained the classifier using multiple instance learning [41] by enlarging the difference of anomaly score for normal and abnormal segments in the video which is gathered from the maximum anomaly score from each normal and abnormal video, respectively. Since normal frames are labeled, the classifier can learn normal patterns easily. The segment which shows the maximum anomaly score in the normal video represents a hard example, and that shows that the maximum anomaly score in the abnormal video can be pseudo-labeled as abnormal frames. However, the pseudo-labeling in an abnormal video using the maximum anomaly score is largely affected by the noise in the video.

To reduce the effect of the noise, Tian et al. [12] sample top-*k* anomaly-scored segments from normal and abnormal videos instead of sampling maximum anomaly-scored segments in Sultani et al.’s work [11]. Tian et al. [12] uses only 1D convolution layers to consider temporal dependency, which is not enough for long-range temporal dependency and is dependent on future input that is not suitable for online anomaly detection in an online fashion. Wu et al. [14] propose a temporal module that can reflect long-range temporal dependency from past input. Furthermore, Sapkota et al. [13] uses a hidden Markov model [42] strategy that handles long-range dependency and can reduce the dependency of parameter *k* in the top-*k* sampling method in Tian et al.’s work [12]. Furthermore, Zhang et al. [15] and Lv et al. [16] also handle the problems of the pseudo-labeling process because the maximum or top-*k* methods cannot fully use all the abnormal frames in the training abnormal video. Zhang et al. [15] estimate the uncertainty of the anomaly detector and perform pseudo-labeling for frames that have the lower uncertainty. Lv et al. [16] find confident normal and abnormal frames using the convergence rate in the training. In addition to pseudo-labeling, the auxiliary task to estimate optical flow or objectiveness can be used to compensate for the lack of labels as in MyeongAh et al.’s work [17]. All these weakly supervised methods require frame-level annotation for normal videos, and video-level labels for abnormal videos.

Unsupervised anomaly detection methods constitute the third category. These methods use both normal and abnormal data in training, albeit without labels indicating whether the frames are normal or abnormal. It is assumed that most of the training data are normal, which is the characteristic of real-world abnormal data. If the reconstructor is trained on the unlabeled dataset, the reconstructor can show the lower reconstruction error not only for normal data, but also for abnormal data. For this reason, most of the method uses the pseudo-labeling process to filter out the small portion of abnormal data.

Yu et al. [18] use self-paced learning [43], which adjusts the learning schedule depending on the difficulty of the task. At the beginning of the training, they give a small weight to the loss function for the hard examples, which is determined by the reconstruction error. As the reconstructor is trained, the weight for hard examples increases accordingly but is still lower than an easy example. According to this procedure, the reconstructor effectively learns normal patterns only. This method performs object-centric anomaly detection, and uses multiple frames and optical flow that result in a significant amount of computation.

Zaheer et al. [1] use features from a deep network [44] pretrained using an action recognition task. They use a reconstructor and a discriminator network. During training, each model generates pseudo-normal and abnormal labels for the other model. The reconstructor is trained to have lower reconstruction errors for pseudo-normal data from the discriminator, whereas the discriminator is trained to discriminate the pseudo-normal and abnormal data from the reconstructor. The pseudo normal and abnormal data are obtained by thresholding the reconstruction error and predicted score. The threshold is obtained using the mean and standard deviation for each batch.

Similar to Zaheer et al. [1], Tur et al. [19] make a pseudo label for normal data using the reconstruction error, but they perform reconstruction using a diffusion model. The input image is corrupted by the noise, and after, the diffusion process model. Zaheer et al. [1] and Tur et al. [19] use deep features from a pre-trained action recognition network which uses about 16 frames for input. The calculation of deep features results in computational cost.

In this study, we follow the unsupervised method in the sense that an unlabeled video dataset is available. In addition to the video dataset, we use a pre-defined set of texts that can describe the normal and abnormal situations of interest. With this simple additional knowledge, we obtain a large performance gain to the unsupervised method. We organize the explanations and limitations of the mostly related works in Table 1.

### 2.2. Large VLMs

Through contrastive training with a large pool of images and captions available online, various large VLMs have been developed. Models pre-trained on large-scale datasets have achieved excellent performance in various downstream tasks. CLIP [23] is a vanilla model that uses separate text and image encoders with a contrastive loss for text and image embeddings. ALIGN [24] is trained using more realistic noisy data than CLIP. BLIP [25] handles the noise of captions on the web by generating and filtering the noisy captions. FLAVA [26] introduces a multi-modal encoder that can boost the performance of multimodal reasoning such as visual question answering (VQA). XVLM [27], FILIP [28], and GLiP [29] use curated region-labeled datasets for contrastive learning between image patches and corresponding texts. We use CLIP as a pre-trained model for feature encoding because it is simple but exhibits satisfactory zero-shot classification performance.

### 2.3. Adaptation of VLMs

Various methods have been developed to adapt VLMs in specific domains. Wortsman et al. [33] proposed a fine-tuning method for a zero-shot classifier using a model ensemble. The authors experimentally demonstrated that the mixing coefficient between the original and fine-tuned model affects the performance gain and generalizability of the fine-tuned zero-shot classifier. Other methods focus on fine-tuning with a limited number of examples [34,35,36,37,38]. Few-shot fine-tuning methods can be divided into two types. The first type is prompt tuning [34,35,36], in which the text input prompt is modified to indirectly modify the text features. The second type uses additional layers to directly modify text or image features [37,38]. All of the methods focus on classification tasks using exact ground truth labels in the dataset. Although our method can be considered as a fine-tuning of a zero-shot classifier, we do not use any image-level labeled dataset. Instead, we train the model in an unsupervised manner. Recently, Menon et al. [22] used multiple prompts from GPT-3 to provide cues for the adaptation of LLMs to the training data domain. However, this approach does not use additional image data for domain-specific adaptation and instead focuses on the general domain.

Large VLMs can be adapted not only to classification tasks but also several other downstream tasks. Researchers have studied the potential of large VLMs in realizing object detection, semantic segmentation, image generation, and other vision tasks [46,47,48,49]. Moreover, these models have been applied to various vision-language tasks, such as VQA, image captioning, phrase grounding, and vision and language navigation [50,51,52]. We use one of the large VLMs, CLIP [23], for video anomaly detection and adapt it to an unlabeled video dataset.

## 3. Proposed Method

This section describes the proposed anomaly detector that uses text descriptions of normal and abnormal situations by calculating the similarity score between the feature vectors of the image and text descriptions using CLIP. The overall framework of the proposed anomaly detector is shown in Figure 2.

In the following subsections, we explain the process of obtaining the text descriptions and inference training of the proposed anomaly detector.

### 3.1. Acquisition of Text Descriptions

Due to the fact that normal and abnormal situations vary depending on the situation of anomaly detection, we aim to obtain texts that cover a wide range of actions or objects. To this end, we use ChatGPT [21] to answer questions such as “Could you provide a list of concise terms that describe both normal and abnormal events commonly recorded by surveillance cameras on <specific location>?”. We substitute the “<specific location>” in the question with a word, such as “campus”, “street”, “grocery store”, “factory”, “house”, “parking lot”, “road”, “park”, “beach”, “mountain”, “building”, “school”, “subway”, and “airport”, depending on specific environments.

Given that ChatGPT is trained on a wide range of texts from the Internet, it can provide text descriptions for normal and abnormal situations that are typical for each location. Using the aforementioned question, we obtain nearly 100 text descriptions for each normal and abnormal situation. We then filter out repetitive descriptions (e.g., “fighting” and “burglary”) and ambiguous descriptions (e.g., “suspicious behavior”) to obtain clear and relevant text descriptions.

Although the text descriptions obtained by ChatGPT are suitable for developing anomaly detectors for the general domain, they may not accurately reflect the normal and abnormal situations of a specific domain. In such cases, one can filter out or add relevant text descriptions to better reflect the domain knowledge. We use UCFcrime [11] to simulate the general domain and ShanghaiTech [31] for the specific domain, i.e., a walking-only zone. We filter out text descriptions of non-walking persons (“running”) or transportation in the ShanghaiTech dataset. Furthermore, we add text descriptions that are used in an object detector. The overall process for obtaining text descriptions is represented in Figure 3. The detailed text descriptions are explained in the experiment section.

### 3.2. Proposed Anomaly Detector Using the Similarity Measure between Image and Text Descriptions

This section explains how we use the CLIP to detect abnormal situations by using the text descriptions described in Section 3.1. Before detecting anomalies, we feed the obtained text descriptions into the text encoder of CLIP to calculate the corresponding normalized text features $f_{i}$, i=1,…,N, where *N* is the number of text descriptions. Since we use the “ViT-B/32” model of CLIP, the dimension of image and text features, $f_{x}$ and $f_{i}$, are both 512-dimensional vectors. For the video input, we calculate the image feature using an image encoder of CLIP. For the UCFcrime dataset [11], we use a single frame as the input. In contrast, for the ShanghaiTech dataset [31], we use the cropped region detected by an object detector, because the object is smaller than the frame.

For each normalized image feature $f_{x}$ extracted from the frames or objects in the input video, one can compute the cosine similarities with pre-calculated normalized text features $f_{i}$. Although this similarity measure can be directly applied to an anomaly detector, the performance in the target domain may not be satisfactory. Thus, we introduce additional trainable parameters to modify the similarity measure of CLIP for the *i*th text description to be adapted to the target domain:(1)$$\begin{array}{c} W\left(f_{x},f_{i}\right)=f_{x}^{T}A_{i}f_{i}/T+b_{i}, \end{array}$$
where $A_{i}\in R^{512\times 512}$ denotes a diagonal matrix; $b_{i}$ denotes a scalar; and *T* is a temperature parameter, which is set as 0.01 in all the experiments. The diagonal elements in $A_{i}$ and $b_{i}$ are trainable parameters that are updated by gradient descent by minimizing the loss in Equation (Equation 10). To make the similarity measure before the update the same as the original cosine similarity, the initial value of the update is the identity matrix and zero for $A_{i}$ and $b_{i}$, respectively. The operation in Equation (Equation 1) can be understood as the weighted inner product of the encoded image feature and the *i*th encoded text description along with the addition of the *i*th bias. The inner product weights for the *i*th encoded text $A_{1\ldots N}$ and bias $b_{1\ldots N}$ are trained to modify the similarity measure. W(·,·) represents the text-conditional similarity between two normalized vectors. Next, the similarities are fed to the SoftMax function and summed over the abnormal text descriptions to obtain the anomaly score s(x) for the frame or object x: (2)$$\begin{array}{c} s\left(x\right)=\sum_{i\in C_{a}}p(c=i|x), \end{array}$$(3)$$\begin{array}{cc} p(c=i|x)= & \frac{\exp\left(W(f_{x},f_{i})\right)}{\sum_{j=1}^{N}\exp(W\left(f_{x},f_{j}\right))}, \end{array}$$
where $C_{a}$ refers to a set of indices corresponding to abnormal text descriptions. The abnormal frame or object is detected when the score exceeds a predetermined threshold.

### 3.3. Training of the Parameters of the Proposed Similarity Measure with Text Descriptions

The proposed anomaly detector uses text-conditional similarity with parameters $A_{1\ldots N},b_{1\ldots N}$. These parameters are trained through the back-propagation of gradients of the two losses defined in the following subsections.

#### 3.3.1. Triplet Loss

Since our purpose is to modify the similarity score, we have to determine which similarity scores with text descriptions should be increased or decreased for each image. There is no labeled video data; we have to use prior knowledge to train the proposed similarity. There are two kinds of prior knowledge: the first one is that most of the video dataset is normal, which is used in the regularization loss in Section 3.3.2. The second prior knowledge is that CLIP classifies the image properly enough, even though there are some miss-classified images. Relying on the performance of CLIP, there are several variations for choosing positive examples, text descriptions whose similarity score should be increased; and negative examples, text descriptions whose similarity score should be decreased. Furthermore, there are variations regarding how to increase or decrease the similarity core with those positive and negative examples.

At first, we have to choose the positive example that has the highest similarity score with the image feature. In the existing CLIP adaptation methods [33,34], the positive sample is the labeled text feature which is the corresponding label for each image. However, we do not have labels, and we choose a positive example as follows:(4)$$\begin{array}{cc} i_{pos} & ={\mathrm{argmax}}_{i}\;p\left(c=i|x_{j}\right), \end{array}$$
where $p(c=i|x_{j})$ is from Equation (Equation 3). One can choose positive examples by top-*k* text descriptions, but we choose top-1 since CLIP’s original classification is performed by the most similar text description.

After choosing the positive example, one can choose the negative example as the rest of the text descriptions, similar to typical cross-entropy loss in the existing CLIP adaptation methods [33,34]. However, their methods apply to the typical classification task where classes are exclusive enough so that it is reasonable to decrease the similarity of the rest of the classes, except for the positive example. Different from those methods, we have a list of text descriptions of which some are not mutually exclusive. For example, the two text descriptions, “person” and “standing”, can describe the image containing a person standing. For this reason, we do not choose negative examples as the rest of the text descriptions. Instead, we choose a negative example from the list of text descriptions whose normality is different from the positive example. This is because the mutual exclusiveness can be much higher when comparing the text descriptions sampled from different normality. Within the text descriptions of different normality with positive examples, we also choose the top-1 negative examples with similar reasons for choosing the top-1 positive example.
(5)$$\begin{array}{cc} i_{neg} & =\left\{\begin{array}{c} {\mathrm{argmax}}_{i\in C_{a}}\;p\left(c=i|x_{j}\right)\;if\;i_{pos}\in C_{n}\\ {\mathrm{argmax}}_{i\in C_{n}}\;p\left(c=i|x_{j}\right),\;\mathrm{otherwise}, \end{array}\right. \end{array}$$
where $C_{n}$ refers to a set of normal text descriptions. After obtaining positive and negative examples for each image, we apply the following triplet loss which is similar to the soft triplet loss defined in [53]:(6)$$\begin{array}{cc} L_{Tri} & =\frac{1}{b}\sum_{j=1}^{b}-\lambda_{j}\log\frac{p(c=i_{pos}|x_{j})}{p\left(c=i_{pos}|x_{j}\right)+p\left(c=i_{neg}|x_{j}\right)} \end{array}$$
where $\lambda_{j}$ is the weight for the loss for each $x_{j}$. To assign more weight to the loss of normal text features, we set $\lambda_{j}$ as 1 when the most similar text description is normal and 0.1 otherwise. This loss can be interpreted as a typical cross-entropy loss with two classes, a positive and a negative example, giving more weight to the class of the positive one.

#### 3.3.2. Regularization Loss

Most of the existing unsupervised methods [1,18] assume that the majority of the video frame data are normal. We also introduce this assumption for the regularization loss to train the parameters. Specifically, we add the regularization loss, which encourages the anomaly detector to assign a lower anomaly score to pseudo-normal data. To obtain the pseudo-normal frames, we apply a threshold to the anomaly scores in the training data and classify data with scores below the threshold as normal. For these pseudo-normal frames, we introduce a loss function that enforces the anomaly score to be lower than a certain value, denoted as α: Examples of a false alarm. Examples showing the effects of training are as follows:(7)$$\begin{array}{cc} L_{Reg}=\frac{1}{b}\sum_{j=1}^{b}L_{Reg}^{j},\;\;\;\;\; & \end{array}$$(8)$$\begin{array}{c} L_{Reg}^{j}=\left\{\begin{array}{cc} \max\left(s(x_{j})-\alpha ,0\right) & if\;s(x_{j})<\tau \\ 0, & \mathrm{otherwise}, \end{array}\right. \end{array}$$

In the experiment, we set τ to be the value that classifies 80% of the data as normal, and α is set as 0.1. After each update of the parameter, we calculate the pseudo-label and loss again.

Furthermore, we introduce the regularization loss that requires the diagonal matrix $A_{i}$ to be an identity matrix:(9)$$\begin{array}{c} L_{Identity}=\sum_{i=1}^{N}{∥A_{i}-I∥}_{2}*\lambda_{Identity}, \end{array}$$
where I is the identity matrix, and $\lambda_{Identity}$ is 0.1 in the experiment. The total loss is defined as follows:(10)$$\begin{array}{c} L_{Total}=L_{Tri}+L_{Reg}+L_{Identity}, \end{array}$$

The overall training process is shown in Figure 4.

## 4. Experiments

The experiments were conducted using an AMD Ryzen 7 3700x CPU which is manufactured by Advanced Micro Devices, Inc. in Santa Clara, CA, United States. and an NVIDIA GeForce RTX 2080Ti GPU manufactured by Nvidia in Santa Clara, CA, United States. We evaluated the performance of the anomaly detector using two metrics: the area under the curve (AUC) of the receiver operating characteristic curve (ROC) and average precision (AP), which is the AUC of the precision–recall curve (PRC). We apply both metrics to all frames at once (Micro) or to each video separately and average them (Macro). The test sets for the ShanghaiTech and UCFcrime datasets have an imbalanced ratio of abnormal frames, i.e., 4.5% and 7.6%, respectively. The AUC yields promising results on imbalanced datasets [14], whereas the AP focuses on the detection of specific labels (abnormal situations, in this study).

The following subsections present details of the datasets, text descriptions, and implementation. Subsequently, we describe the comparison of the proposed method with the existing anomaly detection methods and the conducted ablation studies.

### 4.1. Datasets and Text Descriptions

To simulate the general and specific domains of anomaly detection, we use the UCFcrime [11] and ShanghaiTech [31] datasets, respectively, and assume that the user has some domain knowledge about them. The UCFcrime dataset [11] contains videos captured by surveillance cameras from various locations and labels of 13 categories of abnormal situations, i.e., abuse, arrest, arson, assault, burglary, explosion, fighting, road accidents, robbery, shooting, shoplifting, stealing, and vandalism. Given that the categories in the UCFcrime dataset involve crimes and accidents that threaten public safety and are generally considered abnormal, we use this dataset to simulate the general domain of anomaly detection.

For the UCFcrime dataset, we use the text descriptions for the general domain of anomaly detection, as summarized in Table 2. As discussed in Section 3.1, we obtain these descriptions using ChatGPT and filter them with the help of a user. Although we could have used the 13 abnormal category names in the dataset as the text descriptions, we did not do so because the text descriptions obtained through ChatGPT adequately cover these labels.

The ShanghaiTech dataset [31] consists of videos captured by surveillance cameras placed in various positions within a campus. Although abnormal situations in the general domain of anomaly detection, such as fighting, stealing, and falling down, are also considered abnormal in the dataset, normal situations in the general domain, such as cars, bicycles, motorbikes, and running are excluded from the dataset due to the assumption that the locations are strictly walking-only zones. Consequently, we use the ShanghaiTech dataset to simulate the specific domain of anomaly detection.

According to the definitions of normal and abnormal situations in the ShanghaiTech dataset, we exclude text descriptions containing transportation or running in the general domain, as described in Table 2. Additionally, because we use an object detector trained on the MS COCO dataset [54] following Georgescu et al.’s work [9], we categorize the classes from the object detector into normal and abnormal based on their relevance to the ShanghaiTech dataset. Specifically, we classify the classes of transportation or large animals as abnormal and all other classes as normal. The normal and abnormal text descriptions from the MS COCO dataset are presented in Table 3.

### 4.2. Implementation Details

The following subsections present the implementation details for each dataset and details regarding the temporal smoothing of the anomaly score.

#### 4.2.1. Object Detector in the ShanghaiTech Dataset

Due to the fact that the size of the objects in the ShanghaiTech dataset is smaller than the frames, many existing methods use object detectors before performing anomaly detection [9,55]. Furthermore, abnormal situations in the dataset are object-centric. In other words, abnormal situations that cannot be detected by object detectors, such as smoke or fire, are not present in the dataset. To perform object detection, we use YOLOv3 [32], which was also used in Georgescu et al.’s work [9]. The object detector is pre-trained on the MS COCO dataset [54]. We crop the object with a box larger than the bounding box from the object detector to include the background or context information. The object-level anomaly score is calculated using Equation (Equation 2), and then the maximum value in each frame is used as the frame-level anomaly score.

Unlike the ShanghaiTech dataset, the UCFcrime dataset contains scenes focusing on the object, and several abnormal categories, such as arson and explosion, cannot be detected using object detectors. Thus, we use an image as the input for the anomaly detector without an object detector. Additionally, to avoid redundant frames, we use one frame for every 16 frames in the UCFcrime dataset.

#### 4.2.2. Temporal Difference in UCFcrime Dataset

Data in the UCFcrime dataset are labeled normal when there is no abrupt change in the scene, even if the situation appears abnormal. To account for the dataset’s definition of abnormal situations, we calculate the simple anomaly score using the temporal difference of the image features of CLIP [23]:(11)$$s_{td}\left(x_{t}\right)=\frac{1}{Z}\frac{∥f_{x_{t+1}}-f_{x_{t}}∥}{∥f_{x_{t}}∥},$$
where *Z* is a normalization constant; and $f_{x_{t}}$ and $f_{x_{t+1}}$ are the current and future frame features extracted using the CLIP image encoder, respectively. We normalize each anomaly score s(x) in Equation (Equation 2) and $s_{td}\left(x\right)$ in Equation (Equation 11) to range from 0 to 1 in the test dataset. After normalization, the two scores are multiplied.

#### 4.2.3. Temporal Smoothing of the Anomaly Score

With reference to an existing method [9], we perform the temporal smoothing of the anomaly scores. First, we normalize the anomaly score to range from 0 to 1 in the test dataset. Then, we smooth the score using the 1D Gaussian filter that has also been used in the Georgescu et al.’s work [9].

#### 4.2.4. Various Parameters in Use

We use various parameters for training and inference. We show the parameter used and the reason for these parameters in Table 4. We use *T* in Equation (Equation 1) and $\lambda_{j}$ in Equation (Equation 6) because they show the best performances in both the ShanghaiTech [31] and the UCFcrime [11] datasets. The parameters α in Equation (Equation 8), $\lambda_{Identity}$ in Equation (Equation 9), and τ in Equation (Equation 8) do not make a great difference when changing the values. The σ of 1D Gaussian filter in Section 4.2.3 is the same value as the one used in Georgescu et al.’s work [9]. The ablation study for the parameter value is found in Section 4.5.5 and Section 4.5.6.

### 4.3. Comparison with Existing Anomaly Detection Methods

Table 5 presents a comparison of the proposed method with the existing methods. Due to the fact that the unsupervised methods are rarely applied for anomaly detection, we use the baseline as the one-class classification method from Georgescu et al.’s work [9]. Georgescu et al.’s method [9] is originally trained with a dataset containing normal frames only; however, we train it with an unsupervised dataset that contains mostly normal frames. Although the training dataset contains abnormal frames, Georgescu et al.’s method outperforms the state-of-the-art unsupervised method GCL [1]. The proposed method shows much better results than unsupervised methods, with 8% and 18% better MicroAP metrics, for ShanghaiTech and UCFcrime datasets, respectively. This means that additional information from text descriptions can be used to effectively increase the performance of the anomaly detector. Furthermore, the result shows that the proposed text-conditional similarity effectively adapts CLIP to each video dataset.

Although weakly supervised methods achieve the best performance in the metrics of Micro AUC and AP, they require frame-level anomaly labels for normal videos and video-level labels for abnormal videos, which incurs significant labeling costs. In terms of the remaining metrics, the proposed method outperforms the other methods.

Due to the fact that the AUC metric cannot evaluate the performance of data with only one label, normal videos cannot be evaluated properly using the Macro metric because they contain only normal frames. Thus, the Macro-based metrics can only evaluate abnormal videos. The results indicate that the proposed method outperforms the weakly supervised methods for abnormal videos by 17% and 5% for each dataset.

To verify the result that the weakly supervised method achieves a superior Micro AUC despite its inferior performance on abnormal videos, we compare the anomaly scores of the proposed and weakly supervised methods. We denote $X_{n}$ and $X_{a}$ as a set of frames in the normal and abnormal videos. First, we calculate the mean of the frame-level anomaly scores for normal and abnormal frames in abnormal videos as $s_{\left(frame-N,video-A\right)}$ and $s_{\left(frame-A,video-A\right)}$, respectively. Furthermore, we calculate the mean anomaly score of normal video as $s_{\left(frame-N,video-N\right)}$:(12)$$\begin{array}{c} s_{\left(frame-A,video-A\right)}=\frac{1}{|X_{aa}|}\sum_{x\in X_{aa}}s_{f}\left(x\right),\\ s_{\left(frame-N,video-A\right)}=\frac{1}{|X_{na}|}\sum_{x\in X_{na}}s_{f}\left(x\right),\\ s_{\left(frame-N,video-N\right)}=\frac{1}{|X_{n}|}\sum_{x\in X_{n}}s_{f}\left(x\right),\\ X_{na}=\left\{x_{i}|y_{i}=0\;and\;x_{i}\in X_{a}\right\},\\ X_{aa}=\left\{x_{i}|y_{i}=1\;and\;x_{i}\in X_{a}\right\}, \end{array}$$
where $x_{i}$ and $y_{i}$ represent the frame and its corresponding ground truth label of the anomaly, respectively. $y_{i}$ is 0 when the frame is normal and 1 when it is abnormal. $s_{f}\left(x\right)$ is the estimated frame-level anomaly score for each anomaly detector. If the anomaly detector correctly detects the abnormal situations, the difference in the anomaly scores between the normal and abnormal frames in abnormal videos, $s_{\left(frame-A,video-A\right)}-s_{\left(frame-N,video-A\right)}$, must be larger than the difference in the anomaly scores between the normal frames from the normal and abnormal videos, $s_{\left(frame-N,video-A\right)}-s_{\left(frame-N,video-N\right)}$. The ratio of the two differences is defined as
(13)$$\begin{array}{c} r=\frac{s_{\left(frame-A,video-A\right)}-s_{\left(frame-N,video-A\right)}}{s_{\left(frame-N,video-A\right)}-s_{\left(frame-N,video-N\right)}} \end{array}$$

The ratios of anomaly scores for each method are listed in Table 6. The ratio of the weakly supervised method is higher than those of the other methods, which indicates that weakly supervised methods are prone to discriminate between the normal and abnormal videos, but not between the normal and abnormal frames. This phenomenon is aggravated in the ShanghaiTech dataset. The weakly supervised method cannot effectively discriminate abnormal frames in abnormal videos.

Consequently, in the abnormal videos, the anomaly score of the weakly supervised method is inferior, as shown in Figure 5, which presents the anomaly scores for the Georgescu et al. [9] (unsupervised), RTFM [12] (weakly supervised), and the proposed method over the ShanghaiTech dataset. The score is normalized for each video. The figure shows anomaly scores for two videos. The upper part of the figure shows uniformly sampled frames. The lower part of the figure is the graph of anomaly score, the x-axis is the number of frames, and the y-axis is the anomaly score. The red background in the figure shows GT abnormal frames. There are sampled abnormal objects in the middle of the graph, namely “bicycle” and “falling down”, for each abnormal video. The weakly supervised method cannot effectively identify frames involving abnormal situations. The score visualization results can explain why the Macro AUC or AP of the weakly supervised method is lower than those of the unsupervised and proposed methods. The proposed method achieves more accurate anomaly detection results on abnormal videos than the other methods.

### 4.4. Performance Analysis

We conduct a performance analysis of the proposed anomaly detector, as shown in Table 7. In both the object-centric method and non-object-centric method, the proposed method shows an efficient inference process. All the execution time is obtained in the same aforementioned device. The environment we use is Pytorch 1.10.2 [56] except for the calculation of optical flow, which is computed using Tensorflow 1.15 [57]. The object-centric methods use an object detector and the following anomaly detection process is performed for each object. We set there to be twenty objects in the frame. In non-object-centric methods, GCL and RTFM require significant computations for feature extraction because they use the I3D [58] feature that is extracted from 16 consecutive frames. However, the proposed method for the non-object-centric method uses only one frame for every 16 frames, which can increase the computational efficiency.

### 4.5. Ablation Study

We perform an ablation study pertaining to the text-conditional similarity, the noise of the text descriptions, the assumption of the ratio of abnormal situations in the regularization loss defined in Equation (Equation 8), and each proposed component.

#### 4.5.1. Text-Conditional Similarity

To verify the usefulness of the proposed text-conditional similarity, we compare the proposed method. In the fine-tuning methods of CLIP seen in Section 2.3, prompt-tuning methods modify the text feature indirectly by changing part of the prompt, the input of the text encoder. They require back-pagination of the gradient for the whole text encoder in CLIP when training, whereas we do not. For this reason, we only compare the existing method that directly changes text features without gradient calculation for the text encoder. We use one of the models of WiseFT [33], which directly changes the text feature. We denote the updated text feature by WiseFT as $f_{i}^{WiseFT}$. In training, they use the dot product of image and text features for similarity measure:(14)$$\begin{array}{c} W_{dot}\left(f_{x},f_{i}^{WiseFT}\right)=f_{x}^{T}f_{i}^{WiseFT}/T, \end{array}$$
where $f_{i}^{WiseFT}$ is a trainable vector, which is the same shape as the text feature $f_{i}$ and its initial value is the original text feature $f_{i}$. We set the temperature parameter *T* as 0.01, which is the same as the proposed method. WiseFT originally changes the text feature $f_{i}$ to minimize loss with labeled data. Since we do not have labeled data, we use the same loss $L_{Total}$ as we use, except for the $L_{Identity}$ in Equation (Equation 9), since they do not have parameter $A_{i}$. We use the following loss for training:(15)$$\begin{array}{c} L_{FT}=L_{Tri}^{dot}+L_{Reg}^{dot}+\lambda_{weight}*\sum_{i=1}^{N}∥f_{i}-f_{i}^{FT}∥, \end{array}$$
where $L_{Tri}^{dot}$ and $L_{Reg}^{dot}$ are the modified version of $L_{Tri}$ and $L_{Reg}$ by changing the similarity measure in Equation (Equation 1) to Equation (Equation 14). We set $\lambda_{weight}$ as 0.1. In the inference, they use a model ensemble to calculate the similarity score as follows:(16)$$\begin{array}{c} W_{ensemble}=f_{x}^{T}\left(\left(1-\beta \right)f_{i}+\beta f_{i}^{WiseFT}\right)/T, \end{array}$$
where β is the mixing parameter that is a hyperparameter set by humans. In Table 8, we changed β from 0.2 to 1.0 and denote it as “WiseFT-X”, where X is each value of β. In the table, some metrics in the UCFcrime dataset show inferior performance, but overall, the proposed text-conditional similarity shows better performance. Furthermore, WiseFT requires an additional hyperparameter, and the variance of the performance according to the hyperparameter is rather large. We also verify the text-conditional similarity, especially for the bias term, trained according to the aforementioned motivation in Section 4.5.4.

#### 4.5.2. Noise in the Obtained Text Descriptions

Table 9 shows the results of the ablation study for the proposed method in which 10% of the text descriptions gathered from ChatGPT (Table 2) are randomly omitted to simulate the noise in the obtained text descriptions. Twenty experiments are conducted for each dataset. For the ShanghaiTech dataset, we do not omit the text descriptions from the MS COCO dataset, because we assume that the text descriptions from the object detector can be fully used when performing anomaly detection. The results show that the randomness of both datasets is low, and the proposed method is robust against the small amount of noise in the text description.

#### 4.5.3. Ablation Study for Each Proposed Component

Table 10 presents the result of an ablation study for each proposed module. The symbol “-” indicates that the module is not used. The “Temporal difference” row indicates the use of the temporal difference of the image feature in Equation (Equation 11). The first and second columns show the anomaly detector without any training process, based on the original cosine similarity in CLIP [23]. These two columns compare the effect of the temporal difference, which is more notable in the UCFcrime dataset than in the ShanghaiTech dataset. This phenomenon occurs because the definition of abnormal situations in the UCFcrime dataset focuses on the temporal difference of the scene, as discussed in Section 4.2.2.

The other columns in Table 10 list the performance metrics of the proposed anomaly detector after the training of parameters using different loss functions. For the ShanghaiTech dataset, the triplet loss plays a more significant role in improving the performance. However, in the UCFcrime dataset, both the regularization and triplet losses have similar effects, and there is no merit in using both losses. This result can be explained by the fact that the UCFcrime dataset is more challenging than the ShanghaiTech dataset, and thus, the proposed losses are more effective in the ShanghaiTech dataset. In the last two columns, we present the effect of the temporal difference after training the parameters. As in the case before the training, the temporal difference exerts positive effects on the results for the UCFcrime dataset.

#### 4.5.4. Bias Terms after Training

Figure 6 shows the optimized value of the bias term $\exp(b_{i})$ in Equation (Equation 1) for training with the ShanghaiTech dataset [31]. We choose the largest and smallest 10 biases for each normality. The bias term in text-conditional similarity directly adjusts the similarity score. As we mentioned earlier about the motivation of the proposed similarity, the bias term reflects the frequency of the training dataset. The bias terms for normal descriptions generally have higher values than those for abnormal descriptions. However, each of the normal descriptions is not explainable because certain descriptions do not or rarely occur in the dataset, such as “baseball glove” or “birds flying overhead”. An example of non-explainability is presented in the top-left of Figure 7.

The bias terms for abnormal descriptions can adequately explain the frequency of situations in the video dataset, as shown in Figure 6. Bicycles and motorbikes frequently occur in the ShanghaiTech dataset. The bottom two examples of Figure 7 show the effect or training for abnormal objects.

The smallest bias in abnormal descriptions is “jaywalking”. The jaywalking is typically an abnormal situation in the general domain, and we include it in the abnormal descriptions. However, in the ShanghaiTech dataset, in several situations, people walk along the road outside of the crosswalk, as shown in the upper part of Figure 7. We suppose that the bias term for the description “jaywalking” is trained to achieve a lower similarity score to assign a lower anomaly score.

#### 4.5.5. Assumption of Abnormal Ratio

Figure 8 shows the results of the experiment in which we modify the assumption of the ratio of abnormal data, reflected in the threshold τ in the regularization loss in Equation (Equation 8). Over the ShanghaiTech dataset, the performance degrades slightly (by approximately 0.4%) when the ratio is set as 2%. Overall, the performance of the anomaly detector is not significantly affected by the value of τ. This finding demonstrates the robustness of the proposed method to variations in the ratio of abnormal data. We use 20% as the default abnormal ratio for the regularization loss threshold in other experiments.

#### 4.5.6. Other Various Parameters

Figure 9 shows the results of the experiment in which we change the various parameters in use. The temperature *T* in Equation (Equation 1) and weight for triplet loss for abnormal data $\lambda_{j}$ in Equation (Equation 6) are both important factors in ShanghaiTech dataset [31]. The sensitivity to other parameters α in Equation (Equation 8) and $\lambda_{Identity}$ in Equation (Equation 9) are relatively smaller than *T* and $\lambda_{j}$.

## 5. Conclusions

We develop a simple unsupervised anomaly detection method using text descriptions. First, we obtain text descriptions for different domains of anomaly detection. Specifically, we use one of the large language models, ChatGPT [21], to generate descriptions of typical normal or abnormal situations in the general domain of anomaly detection. For the specific domain of anomaly detection, we modify the text descriptions with domain-specific knowledge to improve their relevance to the domain. After obtaining predefined text descriptions, we classify the input frame or object using the similarity measure between image and text features using one of the large vision and language models, CLIP [23].

To further increase the performance of the anomaly detector, we modify the cosine similarity between the image and text features in CLIP with the proposed text-conditional similarity, which can be used to adapt the CLIP to the unlabeled video dataset. Furthermore, the proposed similarity has some extent of explainability about the occurrence of abnormal situations in the video dataset. We use an unlabeled video dataset to train the parameters of the proposed similarity measure. We use triplet and regularization losses that do not require labels. The results of comprehensive experiments show that the proposed method outperforms unsupervised methods and achieves results comparable to those of the weakly supervised method, which incurs significant costs for labeling the dataset.

However, there are some limitations of the proposed method. The first one is a dependency on the hyperparameters in use. We use various parameters and those parameters should be set by humans, not be optimized automatically. The second drawback is the limitation of explainability. Specifically, in Section 4.5.4, the explainability of the anomaly detector is limited to the abnormal text descriptions. The third limitation is the lack of considering the temporal dependency of abnormal situations. We use no temporal dependency in the ShanghaiTech dataset [31] or a little (two frames) in the UCFcrime dataset [11]. However, some of the real-world abnormal situations can be complicated enough that one or two frames are not enough to capture the abnormal situations.

The future work can be aimed at solving these limitations by making models less dependent on human-controlled parameters, increasing the explainability, and handling the long-range temporal dependency. Furthermore, since we only apply the text descriptions for the unsupervised anomaly detection, the proposed method can be extended to the other scenarios of video dataset, one-class classification, or weakly supervised methods in the future.

## Figures and Tables

**Figure 1 sensors-23-06256-f001:**
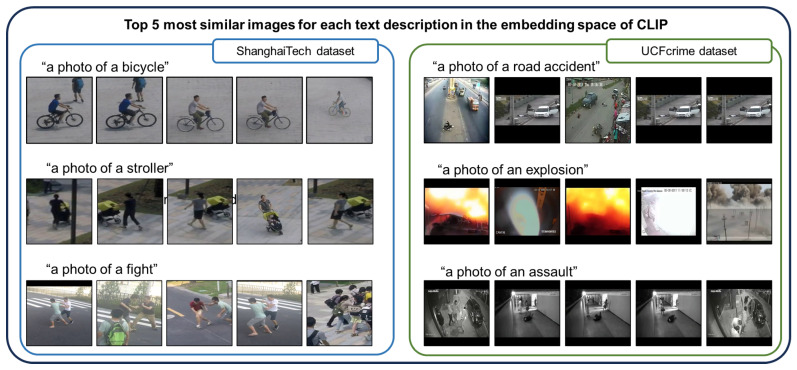
Top five images that are most similar to each text description in both the ShanghaiTech [31] and UCFcrime [11] datasets in the embedding space of CLIP [23]: The similar images for text are obtained by calculating cosine similarity between each image and text feature. Each corresponding image and text feature of CLIP is adequately similar to ensure proper image classification. For the ShanghaiTech dataset [31], we use the cropped image using an object detector [32]. Even for images involving high-level motion (“a photo of a fight” or “a photo of an assault”), the features contain enough information and are similar to the corresponding text descriptions.

**Figure 2 sensors-23-06256-f002:**
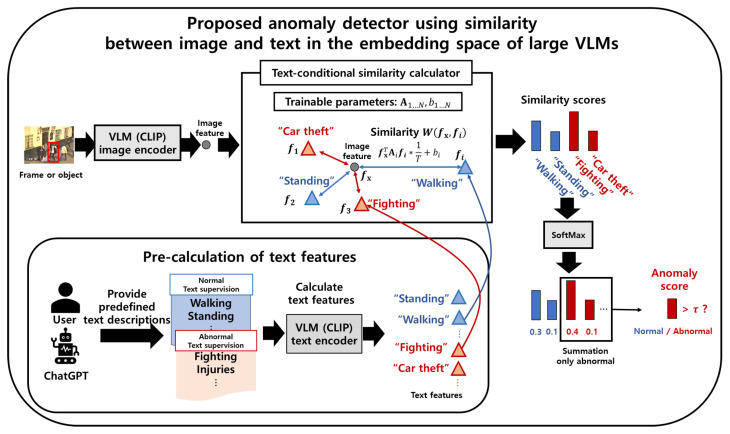
Overall inference process of the proposed anomaly detector using text descriptions: The first row shows the inference process for each frame or object. First, we extract an image feature using the CLIP [23] image encoder. Next, the similarity score between image and text feature vectors is calculated by the proposed text-conditional similarity, which is calculated using Equation (Equation 1). The similarity score is transformed into probabilities using the SoftMax function, and the anomaly score is calculated by summing the abnormal probabilities. If the anomaly score exceeds a certain threshold, the frame or object is classified as abnormal. The lower-left part of the second row shows the process of extracting the text features. The user or ChatGPT provides text descriptions, as described in Section 3.1. The obtained text descriptions are fed to the text encoder of CLIP to extract the text features.

**Figure 3 sensors-23-06256-f003:**
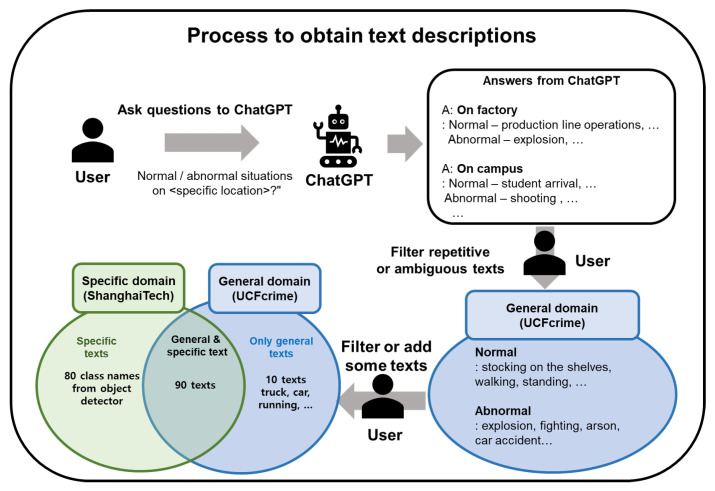
Process to obtain text descriptions: First, a user asks ChatGPT for normal and abnormal situations that commonly occur in various locations. ChatGPT answers the questions, and then the user filters out repetitive or ambiguous descriptions. The obtained text descriptions can be applied in the general domain of anomaly detection (UCFcrime dataset). To apply these text descriptions for a specific domain of anomaly detection, we filter or add descriptions specific to that domain. We use the ShanghaiTech dataset to simulate the specific domain and filter 10 text descriptions from general texts, such as “truck”, “car”, and “running”. Moreover, we add 80 class names from an object detector used only in the ShanghaiTech dataset.

**Figure 4 sensors-23-06256-f004:**
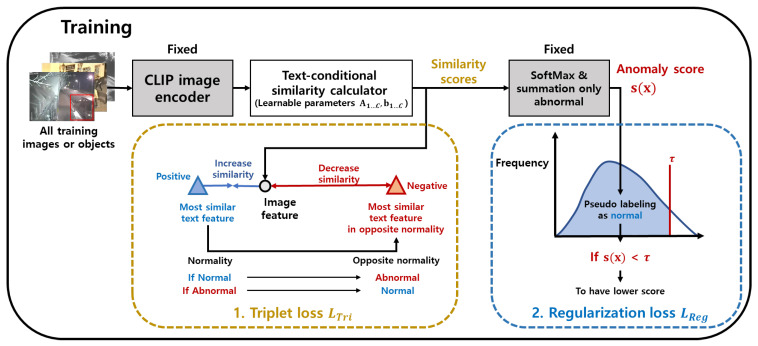
Training of the proposed anomaly detector: Training is performed by back-propagation of the triplet and regularization losses. The bottom-left box shows the triplet loss, i.e., $L_{Tri}$ in Equation (Equation 6), applied to all of the image features by selecting positive and negative examples from the text features. The positive example pertains to the most similar text feature for the image feature. The negative example pertains to the most similar feature among the text features with the normality as different from that of the positive example. We increase the proposed similarity score with the positive example and decrease the score for the negative example. The bottom-right box shows the regularization loss, i.e., $L_{Reg}$ in Equation (Equation 8). According to prior knowledge of the dataset, most of the dataset is normal. The threshold is τ: Image features having lower anomaly scores than τ are assigned lower anomaly scores.

**Figure 5 sensors-23-06256-f005:**
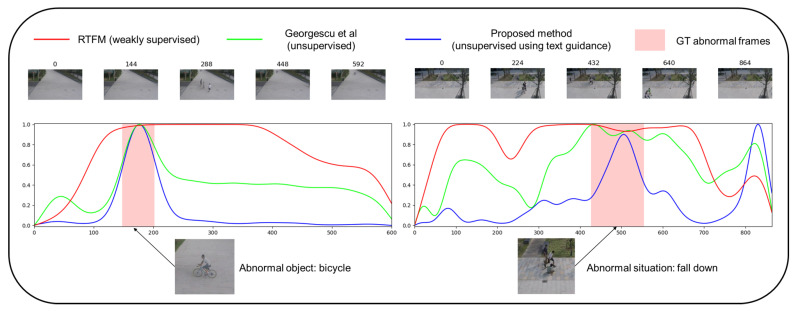
Visualization of anomaly score of the ShanghaiTech dataset: We plot the anomaly scores of the weakly supervised method RTFM [12], unsupervised method from Georgescu et al. [9], and the proposed method. The left graph shows the temporal anomaly scores for the scenario in which an abnormal object (bicycle) emerges. The score shows that the proposed method effectively discriminates between the normal and abnormal situations. The right graph shows scores in which an abnormal situation (falling down) occurs. Although a false alarm occurs after 800 frames, the proposed method outperforms the other methods. The weakly supervised method shows significant false alarms in both graphs and thus exhibits inferior Macro results.

**Figure 6 sensors-23-06256-f006:**
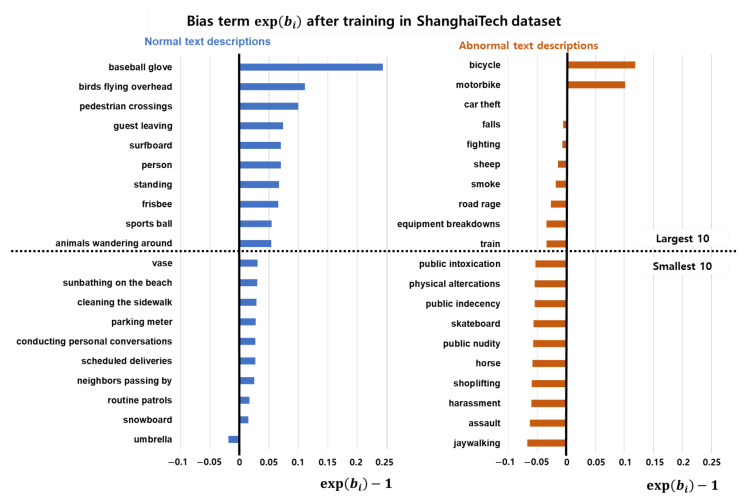
Bias terms of normal and abnormal descriptions after training over the ShanghaiTech dataset [31]: The upper part of the figure shows the largest 10 bias term in each normal and abnormal. text description, and the lower part shows the smallest 10 bias terms. The value is subtracted by 1 to ensure comparison with the results before training when the value is 1. The normal descriptions are not explainable, because the situations described by some of those descriptions are not observed in the video dataset. However, abnormal descriptions are explainable, as the frequently occurring descriptions such as bicycle and motorbikes have larger biases.

**Figure 7 sensors-23-06256-f007:**
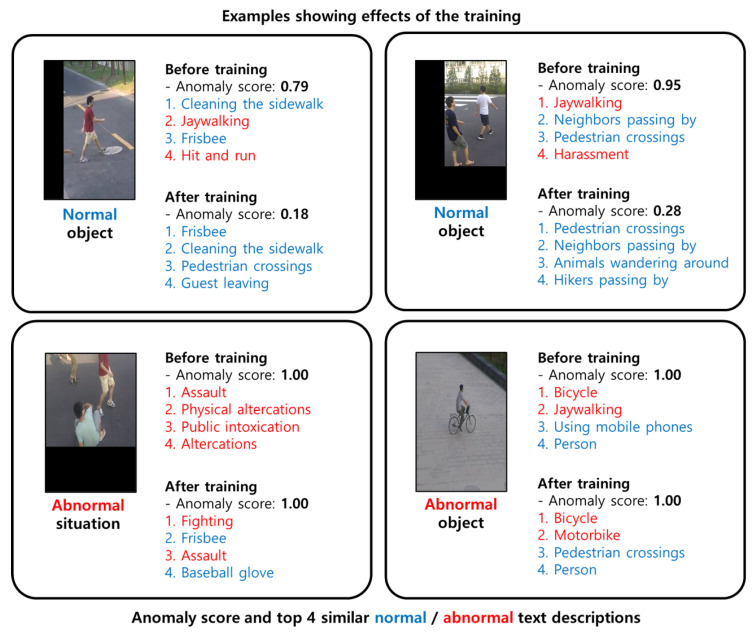
Examples showing the effects of training over the ShanghaiTech dataset [31]: Each example contains objects, the anomaly score is normalized on one video, and the top four most similar text descriptions using the proposed similarity measure are shown. The text descriptions are colored blue and red when they correspond to normal and abnormal text descriptions, respectively. The top-left image shows an example of the decreased explainability of the model. The top-left and right images show that the similarity score with the description “jaywalking” decreases after training because “jaywalking” is considered abnormal in the text descriptions but normal in the video dataset. The two examples at the bottom show the effect on the abnormal object or situation.

**Figure 8 sensors-23-06256-f008:**
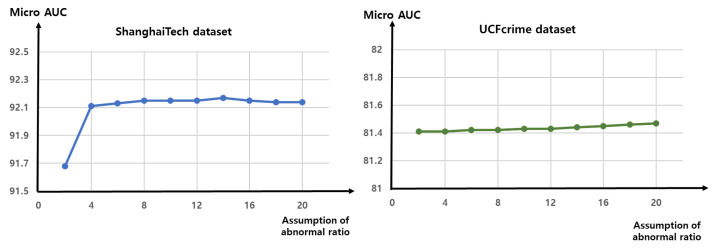
Ablation study in which the τ in the regularization loss in Equation (Equation 8) is changed. We vary τ by changing the assumed ratio of abnormal data in the training dataset. For example, if we assume that 20% of the data are abnormal, then the regularization loss is applied to the remaining 80% of the data, based on the anomaly score. Changing the value of τ does not significantly affect the performance of the anomaly detector, which indicates that the proposed method is robust to different ratios of abnormal data in the training dataset.

**Figure 9 sensors-23-06256-f009:**
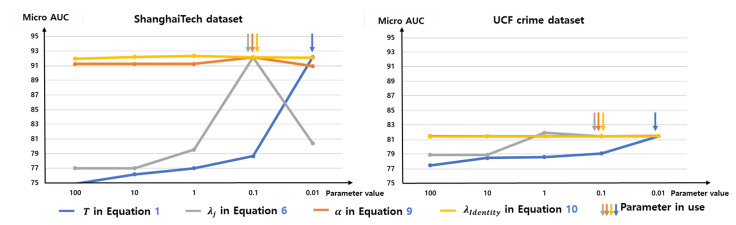
Ablation study for various parameters used in the proposed method. *T* in Equation (Equation 1), $\lambda_{j}$ in Equation (Equation 6), α in Equation (Equation 8), and $\lambda_{Identity}$ in Equation (Equation 9). The arrow shows the value used for the proposed method. The *T* and $\lambda_{j}$ are important factors. Other parameters do not affect performance much.

**Table 1 sensors-23-06256-t001:** Briefexplanations and limitations for most related works with the proposed method.

Method		Explanation	Limitation
OCC	Georgescu et al. [9]	Use pseudo-abnormal data	Require optical flow
	Reiss et al. [10]	*k*NN distance with training data	Cannot be applied to unsupervised dataset
Weakly sup.	RTFM [12]	Pseudo-labeling using top-*k*	Require frame-level labeled normal videos and abnormal videos
	Sapkota et al. [13]	Pseudo-labeling using HDP-HMM	
	Wu et al. [14]	Tightly capture temporal relation	
	Purwanto et al. [45]	Consider relational propagation	
Unsupervised	Yu et al. [18]	Pseudo labeling with self-paced learning	Require a significant of computation for feature extraction using 16 frames
	Zaheer et al. [1]	Pseudo-labeling with cooperative learning	
	Tur et al. [19]	Reconstruction model with diffusion model	

**Table 2 sensors-23-06256-t002:** Normal and abnormal text descriptions from ChatGPT. The last ten descriptions in the normal class (underlined descriptions) are omitted in the ShanghaiTech dataset.

Normal	Abnormal
pedestrian traffic	vandalism
building access	theft
street performers	assault
outdoor dining	trespassing
tourists	property damage
dog walkers	fire
customer browsing	smoke
customer shopping	tornado
stocking shelves	hurricane
employee arrivals	medical emergencies
production line operations	break-in
scheduled deliveries	loitering
pet movement	drug use
neighbors passing by	graffiti
maintenance crew working	public intoxication
cleaning crew working	unruly crowds
pedestrian crossings	shoplifting
children playing	drug dealing
cleaning the sidewalk	hiding merchandise
hikers passing by	altercations
animals wandering around	fighting
birds flying overhead	injuries
sunrise	accidents
sunset	slips
routine patrols	falls
sitting	intruder
standing	equipment malfunctions
reading newspapers	equipment breakdowns
reading books	chemical spills
using mobile phones	chemical leaks
conducting personal conversations	power outages
walking through the station	vehicle accidents
guest arriving	vehicle collisions
guest leaving	physical altercations
cashier scanning	hit and run
cashier bagging items	car theft
restocking shelves	robbery
sunbathing on the beach	road rage
lounging on the beach	jaywalking
buying a ticket	aggressive behavior
delivery trucks	harassment
bicyclists	public indecency
joggers	public nudity
cars driving by	acts of terrorism
emergency vehicles	acts of violence
running	burglary
motorbike	rockslides
people entering parked vehicles	avalanches
traffic flow	dangerous wildlife
vehicle traffic	weapons possession

**Table 3 sensors-23-06256-t003:** Normal and abnormal text descriptions in the MS COCO dataset for the ShanghaiTech dataset.

Normal			Abnormal
person	traffic light	remote	bicycle
fire hydrant	stop sign	mouse	car
parking meter	bench	tvmonitor	motorbike
bird	cat	dog	aeroplane
backpack	umbrella	handbag	bus
tie	suitcase	frisbee	train
skis	snowboard	sports ball	truck
kite	baseball bat	baseball glove	boat
surfboard	tennis racket	bottle	horse
wine glass	cup	fork	sheep
knife	spoon	bowl	cow
banana	apple	sandwich	elephant
orange	broccoli	carrot	bear
hot dog	pizza	donut	zebra
cake	chair	sofa	giraffe
potted plant	bed	dining table	skateboard
toilet			

**Table 4 sensors-23-06256-t004:** The values and corresponding reasons of the various parameters in the proposed method.

Notation	The Value Used in the Proposed Method	Reasons for the Value
*T* in Equation (Equation 1)	0.01	Shows best performance, as shown in Section 4.5.6
$\lambda_{j}$ in Equation (Equation 6)	1 if $i_{pos}\in C_{n}$ 0.1 otherwise $i_{pos}\in C_{a}$	Shows best performance, as shown in Section 4.5.6
α in Equation (Equation 8)	0.1	Does not affect much, as shown in Section 4.5.6
$\lambda_{Identity}$ in Equation (Equation 9)	0.1	Does not affect much, as shown in Section 4.5.6
τ in Equation (Equation 8)	The value that makes 20% of the data to be abnormal by Equation (Equation 8)	Does not affect much, as shown in Section 4.5.5
σ of 1D Gaussian filter in Section 4.2.3	21	Is the same value used in Georgescu et al. [9]

**Table 5 sensors-23-06256-t005:** Comparison of the proposed method with existing unsupervised and weakly supervised anomaly detection methods: The *Micro AUC-AbnVideos* metric indicates the Micro AUC associated with only the videos containing at least one abnormal frame. Due to the fact that the codes for certain methods are not publicly available, the corresponding scores are marked as “-”. The best score for each evaluation metric is boldfaced, and the second-best score is underlined.

		Unsupervised		Unsupervised with text descriptions	Weakly Supervised		
**Dataset**	**Metric**	**GCL [1]**	**Georgescu et al. [9]**	**Proposed** **wrt. unsup./score/wrt. w. sup.**	**RTFM [12]** **(impl./paper)**	**Wu and Liu** **[14]**	**Purwanto et al. [45]**
ShanghaiTech [31]	Micro AUC	78.93	84.77	8%/92.14/−6%	97.39/97.21	**97.48**	96.85
	Micro AP	-	46.64	18%/57.22/−11%	59.63/-	**63.26**	-
	Micro AUC-AbnVideos	-	80.49	5%/**84.50**/17%	70.35/-	68.47	-
	Macro AUC	-	97.21	1%/**97.82**/2%	96.06/-	-	-
	Macro AP	-	97.68	0%/**97.85**/3%	95.15/-	-	-
UCFcrime [11]	Micro AUC	71.04	-	13%/81.27/−5%	83.63/84.30	84.89	**85.00**
	Micro AP	-	-	-/27.91/−11%	24.90/-	**31.10**	-
	Micro AUC-AbnVideos	-	-	-/**67.72**/5%	64.38/-	-	-
	Macro AUC	-	-	-/**85.76**/3%	83.26/-	-	-
	Macro AP	-	-	-/**71.97**/2%	70.32/-	-	-

**Table 6 sensors-23-06256-t006:** The ratio *r* in Equation (Equation 13) of each anomaly detector.

Dataset	Georgescu et al. [9]	Proposed Method	RTFM [12]
ShanghaiTech	0.13	0.13	2.00
UCFcrime	-	1.02	2.52

**Table 7 sensors-23-06256-t007:** Performance analysis of proposed anomaly detection method. Execution times for the inference process of each module in milliseconds. The fastest whole computation time is boldfaced. The percentage shows how faster the proposed method is to other methods. All the methods, except for the optical flow calculation, are measured in the same environment (Pytorch). However, the optical flow calculation for Georgescu et al. is performed on the same PC but using a different library, Tensorflow, because the implementation of Selflow used in Georgescu et al.’s method is only available in Tensorflow.

Object-Centric Method (20 Objects per Frame)					
**Proposed**		**Georgescu et al.** [9]			
CLIP feat. calc. [23]	20.49 ms	Optical flow calc. (Selflow [59])	57.90 ms		
Similarity calc.	0.18 ms	Recon. and classification	4.37 ms		
Total	**20.67 ms (67% faster)**	Total	62.27 ms		
**Non-object-centric method (Frame as input)**					
**Proposed**		**GCL** [1]		**RTFM** [12]	
CLIP feat. calc.	4.79 ms	ResNext [44] feat. calc.	18.79 ms	I3D [58] feat. calc. (10crop)	68.15 ms
Similarity calc.	0.18 ms	Classification	0.11 ms	Classification	2.51 ms
Total	**4.97 ms (74%, 93% faster)**	Total	18.90 ms	Total	70.66 ms

**Table 8 sensors-23-06256-t008:** Effect of text-conditional similarity: We compare the proposed text-conditional similarity with the existing adaptation method, WiseFT [33], with the same losses except for the loss in Equation (Equation 9). The best and second-best methods are boldfaced and underlined, respectively. The existing method [33] requires an additional hyperparameter that regroups the original CLIP and updated model, which is denoted as “X” in “WiseFT-X”. The result shows that some of the existing methods show higher performance in some metrics, but this largely depends on the hyperparameter. Overall, the proposed text-conditional similarity shows the best results.

	Methods	WiseFT-0.2	WiseFT-0.4	WiseFT-0.6	WiseFT-0.8	WiseFT-1.0	Proposed
ShanghaiTech	Micro AUC	87.25	88.54	89.59	91.03	92.12	**92.14**
	Micro AP	45.03	47.43	52.8	54.89	56.07	**57.22**
	Micro AUC-abnVideo	75.75	75.93	78.38	78.31	78.35	**84.50**
	Macro AUC	97.15	97.25	97.26	97.19	97.11	**97.82**
	Macro AP	97.33	97.68	97.78	**97.85**	97.79	**97.85**
UCFcrime	Micro AUC	**81.95**	80.85	79.45	78.90	78.90	81.27
	Micro AP	26.77	24.94	22.03	21.23	23.62	**27.91**
	Micro AUC-abnVideo	67.41	66.89	64.36	63.88	65.61	**67.72**
	Macro AUC	85.58	85.04	84.68	84.27	**86.30**	85.76
	Macro AP	71.98	71.31	71.35	71.14	**72.57**	71.97

**Table 9 sensors-23-06256-t009:** Ablation study with randomly omitted text descriptions: The “w/o omitting” column shows the results using all of the proposed text descriptions. The ”w/omitting” shows the results with 10% of the text descriptions omitted.

	Metric	w/o omitting	w/omitting
ShanghaiTech	Micro AUC	92.14	91.91 ± 1.02
	Macro AUC	97.82	97.70 ± 0.20
UCFcrime	Micro AUC	81.27	81.20 ± 0.81
	Macro AUC	85.76	86.16 ± 1.21

**Table 10 sensors-23-06256-t010:** Ablation study for each proposed component: The best result for each performance metric is boldfaced. The symbols “✓” and “-” in the table mean with and without using the corresponding module, respectively. The “Temporal difference” shows the temporal difference of the image feature in Equation (Equation 11). “Regularization loss” and “Triplet loss” have been defined in Equations (Equation 8) and (Equation 6), respectively.

Module	Temporal difference	-	✓	✓	✓	✓	-
	Regularization loss	-	-	✓	-	✓	✓
	Triplet loss	-	-	-	✓	✓	✓
ShanghaiTech	Micro AUC	76.98	75.04	79.99	90.81	91.58	**92.14**
	Micro AP	19.14	20.19	27.46	58.30	**60.86**	57.22
	Macro AUC	96.43	97.00	96.98	97.60	97.73	**97.82**
	Macro AP	95.61	96.69	96.67	97.68	97.85	**97.85**
UCFcrime	Micro AUC	77.60	78.90	**81.51**	81.47	81.47	78.56
	Micro AP	23.00	23.62	27.88	**27.91**	27.87	23.59
	Macro AUC	83.21	**86.30**	86.01	86.01	86.00	83.02
	Macro AP	69.83	**72.57**	72.25	72.27	72.25	69.82

## Data Availability

Not applicable.

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
