# Peer review of "Unsupervised Video Anomaly Detection Based on Similarity with Predefined Text Descriptions"

_sensors, 2023, doi:10.3390/s23146256_

Round 1

Reviewer 1 Report

1. The abstract of scientific papers should be written the numerical result, which brought to a conclusion. Besides,  performance in terms of percentage must be mentioned in the abstract.

2. There are a lot of citations but they are grouped without detailed information how their contents affect the plot of the presented research. It should be divided and separately described.

3. The authors should clearly describe related work in more detail, contrasting the limitations of the related works. Moreover, the reviewer recommend to ease the overview related works by using overview tables.

4. Some parameters and their values are unknown. It would be better to show all these parameters and explain the reason for those numbers in the table.

5. It needs to highlight the research main contribution with some brief indications and numerical improvement percentages in section of experimental results.

6. The authors could also compare the computation power and complexity of their work with that of the existing approaches.

7. The conclusion and future work part can be extended to have a better understanding of the approach and issues related to that which can be taken into consideration for future work.

8. It is my understanding that a lot of works have been done in studying the related issues, such as [A]. To put their work into proper perspective, the authors should consider recently published leading-edge articles in the field, preferably from top journals and conferences.

[A] X. Wang, Z. Che, B. Jiang, N. Xiao, K. Yang, J. Tang, J. Ye, J. Wang and Q. Qi, Robust Unsupervised Video Anomaly Detection by Multi-Path Frame Prediction, IEEE Transactions on Neural Networks and Learning Systems, 2020.

Author Response

We sincerely appreciate your valuable comments. 

Reviewer 2 Report

This work proposes a new similarity measure named "text-conditional similarity" to take advantage of the text labels generated by vision and language models (VLM) for improving anomalies in videos.  Conceptually, this appears to be most closely related to the human action recognition work cited on line 86 near the end of the introduction section.  Since human action recognition is the main concept, I'd like to see this concept to be introduced much earlier in the introduction section.

Since the "text-conditional similarity" is the main contribution of the work, it is surprising to not see it in the bullets listed as main contributions of this work (starting on line 101).

The first appearance of the term "text-conditional similarity" on line 74 simply describe it as a modification of the cosine similarity of CLIP, which is very unsatisfying because it does not say why it might be useful to do this modification.  Typically, one would provide some motivation for the proposed changes.

Another important concept appears to be the use of "triplet loss," however, it is never explained why it is something useful to do.  Even the section 3.3.1 does not say anything about how is this "triplet loss" is different from the typical loss function used.

I don't quite see how the experiments demonstrate the usefulness of the "text-conditional similarity."  In general, it would be useful for the experiments to test whether the proposed new ideas achieve their intended objectives.

In Section 3.2, the sizes of f_x, f_i and A_i are not given.  In addition, The authors only state that A_i is a diagonal matrix with given any procedure to determine the diagonal values.

Is it appropriate to add up the anomaly score assigned to different abnormal categories?  Does it make sense to add up possibility of "jaywalking" with "rockslide"?

The conclusion does not talk about the main concepts such as "text-conditional similarity" that were initially described as key concepts introduced in this work.

The second sentence of the introduction dives into technical details that are not very helpful as this stage of the introduction, the opening sentences of the abstract are much more appropriate for this context.

The introduction section touches on several important concepts such as “high-level motion” “motion-containing objects”, “objects” “similarity,” but does not provide explanation or definition.

Author Response

We sincerely appreciate your valuable comments.
Please see the attachment.

Reviewer 3 Report

The proposed method outperforms unsupervised methods and exhibits a performance comparable to weakly supervised methods that require resource-intensive dataset labeling. The work is very well presented. Well cited and well written. I would encourage to accept this paper. That would provide a new way to the scientific community.  

I have one minor concern, please have look grammars. 

Author Response

(The authors gave the same response as above.)

Reviewer 4 Report

The article is devoted to solving the problem of detecting anomalies in video. The topic of the article is relevant. The structure of the article is not classical for MDPI (Introduction, Models and Methods, Experiments, Discussion, Conclusions). The level of English is acceptable. The article is easy to read. The quality of the figures is low. The article cites 75 sources, many of which are not relevant. The References section is sloppy.

The following comments and recommendations can be formulated on the material of the article:

1. I must say right away that the declared original idea must be properly substantiated. It is necessary to clearly and mathematically define what a “text anomaly on video” is. For example, a camera shoots a store sign from the window of a moving car. The view of the inscription will change all the time, but there will be no anomaly here. If a few frames are missing, then this is also not a problem. Anyway, text recognition systems in the view work with frames. If in many frames among the recognized name "Uncle Joe's" on one of the frames there is "Aunt Mary's", is this an anomaly? How to separate the natural text distortion from the artificial one? This is the fundamental question in this task.

2. The task of text recognition (and hence its detection) is very popular, so there are ready-made solutions TextBoxes ++ (Caffe) and SegLinks, but EAST, in my opinion, is the simplest and most accessible. The idea of the EAST detector is to predict not the coordinates of the corners of bounding boxes (for texts or individual letters), but the following three things: - Text Score Maps (probability of finding text in each pixel); - Rotation angle of each box; - Distances to the borders of the rectangle for each pixel. Thus, it is more like a segmentation task (selection of text masks) than a detection task. Here is also a great opportunity for anomalies to squeak, if some letters are rotated to the “wrong” angle relative to the main text - something is wrong here.

3. After detecting the text, I want to immediately feed it to another neural network in order to recognize it and give out a character string. Here you can notice an interesting change of modality - from pictures to text. You should not be afraid of this at all, because everything depends only on what the network architecture is, what exactly is predicted on the last layer and what loss is used. For example, MORAN (PyTorch code) and ASTER (TensorFlow code) do the job quite well. There is nothing supernatural in them, however, two fundamentally different types of neural networks are used very competently at once: CNN and RNN. The first is needed to extract features from the picture, and the second to generate text. Depending on the recognition result, it is also possible to draw conclusions regarding anomalies. There is a wide field for setting the research problem.

4. However, despite the rotated boxes from EAST, recognition networks still receive a rectangular image as input, which means that the text inside it may not occupy all the space. In order to make it easier for the recognizer to predict the text on it directly from the picture, you can transform it in a certain way. We can apply an affine transform to the input image to stretch/rotate the text. This can be achieved using the Spatial Transformet Network (STN), since it independently learns such transformations and is easily integrated into other neural networks (by the way, you can do this alignment for any picture, not only for text). The detection of anomalies in the STN results is also very good and interesting.

5. In pursuit of point 4. MORAN (the same neural network for text recognition) is even smarter - it is not limited to the family of affine transformations, but predicts for each pixel of the input image a map of displacements in x and y, thus achieving any transformation, which will improve network training for recognition. This method is called rectification, that is, correcting the image using an auxiliary neural network (rectifier). Looking for anomalies in such a map is a pleasure.

-

Author Response

(The authors gave the same response as above.)

Round 2

Reviewer 1 Report

This paper has edited and revised according to the reviewer's suggestions.

Reviewer 4 Report

I formulated the following remarks to the basic version of the article:

1. I must say right away that the declared original idea must be properly substantiated. It is necessary to clearly and mathematically define what a “text anomaly on video” is. For example, a camera shoots a store sign from the window of a moving car. The view of the inscription will change all the time, but there will be no anomaly here. If a few frames are missing, then this is also not a problem. Anyway, text recognition systems in the view work with frames. If in many frames among the recognized name "Uncle Joe's" on one of the frames there is "Aunt Mary's", is this an anomaly? How to separate the natural text distortion from the artificial one? This is the fundamental question in this task.

2. The task of text recognition (and hence its detection) is very popular, so there are ready-made solutions TextBoxes ++ (Caffe) and SegLinks, but EAST, in my opinion, is the simplest and most accessible. The idea of the EAST detector is to predict not the coordinates of the corners of bounding boxes (for texts or individual letters), but the following three things: - Text Score Maps (probability of finding text in each pixel); - Rotation angle of each box; - Distances to the borders of the rectangle for each pixel. Thus, it is more like a segmentation task (selection of text masks) than a detection task. Here is also a great opportunity for anomalies to squeak, if some letters are rotated to the “wrong” angle relative to the main text - something is wrong here.

3. After detecting the text, I want to immediately feed it to another neural network in order to recognize it and give out a character string. Here you can notice an interesting change of modality - from pictures to text. You should not be afraid of this at all, because everything depends only on what the network architecture is, what exactly is predicted on the last layer and what loss is used. For example, MORAN (PyTorch code) and ASTER (TensorFlow code) do the job quite well. There is nothing supernatural in them, however, two fundamentally different types of neural networks are used very competently at once: CNN and RNN. The first is needed to extract features from the picture, and the second to generate text. Depending on the recognition result, it is also possible to draw conclusions regarding anomalies. There is a wide field for setting the research problem.

4. However, despite the rotated boxes from EAST, recognition networks still receive a rectangular image as input, which means that the text inside it may not occupy all the space. In order to make it easier for the recognizer to predict the text on it directly from the picture, you can transform it in a certain way. We can apply an affine transform to the input image to stretch/rotate the text. This can be achieved using the Spatial Transformet Network (STN), since it independently learns such transformations and is easily integrated into other neural networks (by the way, you can do this alignment for any picture, not only for text). The detection of anomalies in the STN results is also very good and interesting.

5. In pursuit of point 4. MORAN (the same neural network for text recognition) is even smarter - it is not limited to the family of affine transformations, but predicts for each pixel of the input image a map of displacements in x and y, thus achieving any transformation, which will improve network training for recognition. This method is called rectification, that is, correcting the image using an auxiliary neural network (rectifier). Looking for anomalies in such a map is a pleasure.

The authors responded to my comments. I can't say that their answers are deep enough. However, they correspond to the level of the article. In this context, I recommend the current version of the article for publication. I wish the authors creative success.

-